# Evaluation of a health information exchange system for microcephaly case-finding — New York City, 2013—2015

**Eugenie Poirot**[1,2], **Carrie W. Mills**[2], **Andrew D. Fair**[2,3], **Krishika A. Graham**[2],
**Emily Martinez**[2], **Lauren Schreibstein**[3], **Achala Talati**[2], **Katharine H. McVeigh**[2]*

**1** Epidemic Intelligence Service, Division of Scientific Education and Professional Development, Centers for Disease Control and Prevention, Atlanta, Georgia, United States of America, **2** New York City Department of Health and Mental Hygiene, New York, New York, United States of America, **3** Bronx RHIO, New York, New York, United States of America

* tmcveigh@health.nyc.gov

## Abstract

### Background

Birth defects surveillance in the United States is conducted principally by review of routine but lagged reporting to statewide congenital malformations registries of diagnoses by hospitals or other health care providers, a process that is not designed to rapidly detect changes in prevalence. Health information exchange (HIE) systems are well suited for rapid surveillance, but information is limited about their effectiveness at detecting birth defects. We evaluated HIE data to detect microcephaly diagnosed at birth during January 1, 2013–December 31, 2015 before known introduction of Zika virus in North America.

### Methods

Data from an HIE system were queried for microcephaly diagnostic codes on day of birth or during the first two days after birth at three Bronx hospitals for births to New York City resident mothers. Suspected cases identified by HIE data were compared with microcephaly cases that had been identified through direct inquiry of hospital records and confirmed by chart abstraction in a previous study of the same cohort.

### Results

Of 16,910 live births, 43 suspected microcephaly cases were identified through an HIE system compared to 67 confirmed cases that had been identified as part of the prior study. A total of 39 confirmed cases were found by both studies (sensitivity = 58.21%, 95% CI: 45.52–70.15%; positive predictive value = 90.70%, 95% CI: 77.86–97.41%; negative predictive value = 99.83%, 95% CI: 99.76–99.89% for HIE data).

### Conclusion

Despite limitations, HIE systems could be used for rapid newborn microcephaly surveillance, especially in the many jurisdictions where more labor-intensive approaches are not

**Data Availability Statement:** The authors have made available all the data required to replicate the analysis described in this manuscript. The data can be found in Table 1. For readers interested in the

data used in the New York State Retrospective Chart Review study [Graham et al. 2017] and information abstracted from charts by trained clinicians, readers can contact that study's corresponding authors Deborah J. Fox (deb. fox@health.nyc.gov) and Krishika A. Graham (kgraham1@health.nyc.gov) for data requests. Readers interested in the data obtained from the Bronx RHIO can contact their Chief Operating Officer, Kathryn Miller (kmiller@bronxrhio.org) for data requests.

**Funding:** The authors received no specific funding for this work. However, the work was framed by the Surveillance, Intervention, and Referral to Services Activities for Infants with Microcephaly or Other Adverse Outcomes linked with the Zika Virus - High Risk Local Areas grant (cooperative agreement NU50DD000044) awarded to NYC DOHMH from the U.S. Centers for Disease Control and Prevention (https://www.cdc.gov/pregnancy/zika/research/birth-defects.html) and grant funds supported obtaining and abstracting 4 medical records. The contents expressed by the authors contributing to this work do not necessarily reflect the opinions of the CDC or the institutions with which the authors are affiliated.

**Competing interests:** The authors have declared that no competing interests exist.

feasible. Future work is needed to improve electronic medical record documentation quality to improve sensitivity and reduce misclassification.

## Introduction

Birth defects surveillance in the United States has historically relied on routine reporting of individual cases from hospitals and other health care providers to congenital malformations registries (CMRs) in accordance with state public health laws and regulations [1]. The spread of Zika virus in early 2015 in the Region of the Americas clarified the limitations of this approach for identifying birth defects potentially linked to Zika virus infection, including microcephaly. In response, the U.S. Centers for Disease Control and Prevention (CDC) awarded funding to jurisdictions, including New York City (NYC), to support rapid, active surveillance of Zika-related birth defects [2]. In early 2017, as part of the federal funding awarded to jurisdictions, the NYC Department of Health and Mental Hygiene established Zika-related Birth Defects Surveillance. To operationalize "rapid" reporting, trained surveillance staff sent requests to birthing facilities to submit lists of newborns with diagnosis codes consistent with microcephaly and other Zika-related birth defects from electronic medical record (EMR) systems on a quarterly basis. Surveillance staff then requested, reviewed and abstracted the records on each list ("active" surveillance). Continued funding of these rapid and active surveillance components has been discontinued or scaled back in many jurisdictions and the use of new and supplemental data sources for ascertaining birth defects cases are needed.

Health information exchange (HIE) systems facilitate the transfer of timely and detailed electronic data or information, including clinical data from providers, health insurance claims history, and public health data (e.g., immunization registries) across disparate healthcare information systems involved in the delivery of care. HIE systems present opportunities to advance disease surveillance and have been linked to improvements in child and adolescent immunization status [3], timeliness of notifiable disease reporting [4], reduction of duplicative diagnostic testing and identification of drug seeking behaviors [5], and improved identification of high utilizing vulnerable patients returning within 72 hours of initial emergency department discharge [6].

New York City is one of a growing number of jurisdictions to have partnerships with HIE systems to facilitate exchange of clinical patient information across healthcare organizations. To identify additional ways to support rapid surveillance of birth defects, the NYC Department of Health and Mental Hygiene evaluated data from an HIE system to detect microcephaly diagnosed at birth during 2013–2015, before known introduction of Zika virus in North America. Suspected cases identified by HIE data were compared with cases identified and confirmed to meet the case definition for microcephaly in a prior study (the New York State Retrospective Chart Review, NYSRCR). The objective of this analysis was to determine the extent to which querying of HIE data could replicate the yield obtained by the NYSRCR study through more labor-intensive and costly approaches involving direct inquiry of hospital records and chart abstraction. The sensitivity, specificity, positive and negative predictive values of querying HIE data for identifying confirmed cases of microcephaly were estimated.

## Methods

Data from an HIE system, covering the majority of healthcare provided to the 1.4 million residents in the borough of the Bronx, were queried for *International Classification of Diseases*,

*Ninth Revision*, *Clinical Modification* (ICD-9-CM) and *International Classification of Diseases*, *Tenth Revision*, *Clinical Modification* (ICD-10-CM) microcephaly diagnostic codes (ICD-9-CM code 742.1, ICD-10-CM code Q02) on day of birth or during the first two days after birth at three Bronx hospitals for births to NYC resident mothers occurring January 1, 2013–December 31, 2015. All data from the HIE are housed in a store-based system that uses search engine technology to query records. Records were retrieved for all births with a microcephaly diagnostic code in any care setting, and then limited to births with diagnoses occurring 0–2 days from the date of birth. Python programming language was then used to extract pertinent data from medical records. Data on selected elements, including hospital name, medical record number, date of diagnosis, visit type, and date of birth, were extracted in csv format and exported for analysis. During this period, the HIE system had clinical data on 1,538,602 unique patients, with 1,113,852 (78%) reporting a Bronx ZIP code. Cases identified by HIE were classified as suspected cases pending confirmation of the presence of microcephaly. One suspected case born to a non-NYC resident mother was excluded from the analysis.

Suspected cases identified by querying the HIE system were compared with confirmed microcephaly cases previously identified in the NYSRCR study. In that study, which occurred during the Spring of 2016, the New York CMR asked all New York birth hospitals to query their EMR systems to identify newborns with ICD-9-CM and ICD-10-CM codes specifying microcephaly (ICD-9-CM code 742.1, ICD-10-CM code Q02) for the 2013–2015 period. Additional cases were identified through review of diagnoses recorded in hospital discharge administrative data. Medical records for suspected microcephaly cases were obtained, and microcephaly diagnosis was confirmed on chart review by trained clinicians [7] using the case definition for overall microcephaly developed by the CDC and the National Birth Defects Prevention Network [8]. Some suspected cases did not meet the overall microcephaly case definition and were excluded either because they were misclassified (e.g., macrocephaly, microcephalus), or because both a physician diagnosis and anthropometric information needed to accurately classify head circumference percentile were missing [7]. Medical chart review methods such as this are the gold standard for evaluating the data quality of birth defects surveillance systems [9].

Cases born to NYC resident mothers were matched for three birth hospitals covered by both data sources on birth hospital name, date of birth, and medical record number. Records were linked using a deterministic approach. Cases were classified as a match if the two records agreed on all identifiers and a nonmatch if the two records disagreed on any of the identifiers. Sensitivity, specificity, positive predictive value, negative predictive value, and accuracy of HIE data in identifying confirmed cases of microcephaly were calculated and corresponding Clopper-Pearson exact 95% confidence intervals were estimated. Newborn records for suspected cases identified using HIE data but not by NYSRCR were requested and reviewed by trained clinical abstractors to confirm the presence of microcephaly.

We compared the HIE data to data obtained by NYSRCR across multiple dimensions. Using NYSRCR data as the denominator, sensitivity measured the proportion of confirmed microcephaly cases identified by querying the HIE system. Conversely, specificity measured the proportion of non-cases identified. Using the HIE data as the denominator, the positive predictive value and negative predictive value quantified the accuracy of the HIE data classification of suspected cases and non-cases, respectively. The rate of false positives was calculated as the proportion of cases identified by the HIE query with no match in NYSRCR using the total number of suspected cases identified by the HIE query as the denominator. Lastly, to measure the correspondence between the two systems (i.e., the level of agreement that could be expected by chance, based on the marginal frequencies in both systems), a kappa statistic

**Table 1.  Microcephaly cases born to NYC resident mothers and diagnosed at ages 0–2 days at three birth hospitals—New York City, January 1, 2013 –December 31, 2015.**

| | | Confirmed case identified through direct inquiry to hospitals | | |
| --- | --- | --- | --- | --- |
| | | Yes | No | Total |
| **Suspected case identified in HIE system** | Yes | 39 | 4[a] | 43 |
| | No | 28 | 16,839 | 16,867 |
| | Total | 67 | 16,843 | 16,910 |

HIE, health information exchange.

[a]Records were requested and reviewed by trained clinical abstractors to confirm the presence of microcephaly. One of 4 cases was found to meet the microcephaly case definition.

was calculated. All statistical analyses were performed using SAS 9.4 (SAS Institute Inc, Cary, North Carolina).

## Results

Forty-three suspected cases of microcephaly were identified through the HIE system and 67 confirmed cases were identified by NYSRCR (Table 1). Overall, the HIE system correctly classified birth records 99.81% (16,878/16,910) of the time (Kappa = 0.71). Thirty-nine cases were found in both systems, resulting in sensitivity of 58.21 (39/67), and specificity of 99.98% (16,839/16,843) (Table 2). The overall positive predictive value for HIE data was 90.70% (39/43). Of the four suspected cases identified by the HIE system but missed by NYSRCR, three were misclassified and one was found to meet the microcephaly case definition after chart review, yielding a 7.00% (3/43) false positive rate. No reason was identified for the missed confirmed case.

## Discussion and conclusion

This analysis sought to evaluate the use of querying HIE data to replace or enhance more labor-intensive microcephaly surveillance approaches involving direct inquiry to hospitals and abstraction of hospital records. Leveraging data from an existing HIE system demonstrated high specificity with few false positives when used to detect cases of microcephaly in three

**Table 2.  Evaluation of sensitivity, specificity, and positive and negative predictive values of health information exchange data in identifying confirmed cases of microcephaly—New York City, January 1, 2013–December 31, 2015[a].**

| Measure | |
| --- | --- |
| Sensitivity, % (95% CI) | 58.21 (45.52–70.15) |
| Specificity, % (95% CI) | 99.98 (99.94–99.99) |
| Positive predictive value, % (95% CI) | 90.70 (77.86–97.41) |
| Negative predictive value, % (95% CI) | 99.83 (99.76–99.89) |
| Accuracy, % (95% CI) | 99.81 (99.73–99.87) |
| Total false positives, % (n) | 7.00 (3) |

CI, confidence interval.

[a]Measures of sensitivity, specificity, positive predictive value, negative predictive value, accuracy, and total false positives were calculated using microcephaly cases identified through direct inquiry of hospitals followed by medical chart review for comparisons across the two data sources.

NYC birth hospitals. Querying HIE data identified over half (39/67 = 58%) of the cases; however, approximately 42% (28/67) of confirmed microcephaly cases were missed, potentially because of lack of consistent and complete data documentation within the EMR. In addition, three non-cases (7%) were identified, which is consistent with the 6% of suspected cases that did not meet the microcephaly case definition in the gold standard NYSRCR study. We found no explanation for the one case that was identified by HIE query but missed by NYSRCR, but hypothesize that the microcephaly diagnostic code was recorded in a secondary field that was picked up by the HIE, but not by the reporting hospital's query in the NYSRCR study.

These findings suggest that HIE systems could support the continuation of rapid newborn microcephaly surveillance through near real-time monitoring of clinical data without the burden of managing multiple healthcare settings and systems. The burden on public health departments to request lists of suspected cases, and on hospitals to respond can be high, and may not be cost effective in the absence of an outbreak. Furthermore, in NYC, when using direct inquiry for Zika-related Birth Defects Surveillance, the reporting lag time from birth to issuing requests to hospitals ranged from 0–31 days plus another 28–35 days for receipt of a list of suspected cases, a longer surveillance cycle than could be achieved with an automated monthly HIE report. HIEs might be especially useful for programs that will not continue using rapid case-finding methods initiated in response to the Zika virus outbreaks, or for those that exclusively rely on traditional CMR reporting. Automated monthly querying of HIE systems could be a timely and cost-effective approach to monitor trends in prevalence of microcephaly cases and to detect potential disease outbreaks related to birth defects. Alternatively, where routine direct inquiry is warranted, supplementing existing surveillance methods with timely and automated HIE queries could make systems that rely on hospital inquiry and confirmatory chart review more efficient. Cases identified by both systems could bypass human review so that limited resources could be devoted to investigating cases that are only picked up by one source. For example, the highly specific HIE list of suspected cases could be compared to cases identified by direct inquiry to prioritize cases requiring confirmatory chart review. Using HIE data in this way may also have applications for monitoring other clinical conditions and emerging health threats.

The findings in this report are subject to at least three limitations. First, the detection of cases using HIE data was limited to microcephaly diagnoses queried using a single ICD-CM-9 or ICD-CM-10 code on day of birth or during the first two days after birth. Diagnoses entered into the newborn record later were not captured. Additionally, unlike the NYSRCR, we were not able to classify cases by severity. Accurate identification of complex diagnoses, such as severe microcephaly, requires development of novel detection algorithms that utilize clinically detailed information because diagnostic codes alone (e.g., ICD-CM codes) are limited by suboptimal accuracy in identifying specific birth defects [10, 11]. Microcephaly is a clinical and anthropometrical sign that can be multifactorial with a spectrum of clinical manifestations, making its diagnosis challenging [12]. As natural language processing of EMR data improves it may be technologically possible to capture or index clinical notes related to head circumference, for example, but even so that information may not be included in HIE data exchange.

Second, sensitivity was low, suggesting that HIE data would need to be supplemented with direct inquiry of hospitals. An alternative approach could involve conducting confirmatory review of any charts that were not identified by both systems. Differences in reporting birth defects across participating facilities may be responsible for the under-ascertainment of cases we observed. Identifying infants with microcephaly can be challenging because of differences in clinical case definitions, timing and setting of diagnosis, and case methods [13]. In fact, despite the increasing adoption and implementation of EMR systems, there has been a lack of standardization in design and structure, as well as in adoption of documentation workflows.

Data in these systems have multiple fields for reporting conceptually related information that can vary in format (e.g., as free text or as a diagnostic code). Variation in documentation practices across the three NYC birth hospitals in the HIE system could explain why cases were missed by querying HIE data. Different hospitals may have different workflows in place for coding data and sending information to the HIE. Some hospitals may consistently enter certain data elements (like microcephaly diagnoses) at the point of care whereas other hospitals may be more likely to enter these data elements as post-coded updates upon review after discharge. Depending on how the data gets entered in the EMR, it may trigger a delayed update to the HIE or none at all. These variations call for better standardization of reporting and HIE practices across EMR systems. Continued effort to develop standards for EMR design that will produce clearer documentation of the clinical workflow and requirements for data capture could lessen these challenges and improve how EMR systems can be used for public health surveillance and assist health departments. Furthermore, this analysis queried HIE data only for microcephaly diagnostic codes, but it is possible that clinical signs of microcephaly may appear in the HIE in other forms such as transcribed notes and radiology reports. Future studies could explore the use of natural language processing to parse out clinical signs of microcephaly from unstructured or semi-structured text data.

Third, this analysis only includes cases from three birth hospitals in the HIE system in the Bronx, covering roughly 38% (16,910/44,260) of all births born to NYC resident mothers in the Bronx, which excludes cases from other birth hospitals in the region [14]. This limits the generalizability of our results. Jurisdictions considering the use of HIE data to monitor trends in microcephaly or Zika-related birth defects more generally will need to evaluate the sensitivity and specificity of HIE reporting relative to cases identified by their current Zika-related Birth Defects Surveillance systems. Conducting chart review validation studies over time to assess changes in HIE sensitivity and specificity can provide a framework for evaluating the performance of HIE data; activities to improve documentation of congenital microcephaly within EMR systems may occur concurrently.

Given the continued potential for Zika virus exposures by nonimmune women of childbearing age and financial feasibility for jurisdictions to periodically run HIE data queries, jurisdictions may consider collaborating with a local HIE for monitoring ongoing trends of Zika-related birth defects. HIE data can be used to facilitate the rapid collection of critical clinical information but improvements in documentation and reporting practices are needed.

## Acknowledgments

We wish to thank the Congenital Malformations Registry, New York State Department of Health, the Primary Care Information Project, New York City Department of Health and Mental Hygiene, the Division of Prevention and Primary Care, Jocelyn Chacko, Mary Crippen, Dr. Hannah Gould, and Tenzin Tseyang.

## Author Contributions

**Conceptualization:** Eugenie Poirot, Katharine H. McVeigh.

**Formal analysis:** Eugenie Poirot, Carrie W. Mills, Andrew D. Fair, Emily Martinez, Lauren Schreibstein, Katharine H. McVeigh.

**Methodology:** Eugenie Poirot, Carrie W. Mills, Andrew D. Fair, Krishika A. Graham, Emily Martinez, Lauren Schreibstein, Achala Talati, Katharine H. McVeigh.

**Project administration:** Eugenie Poirot.

**Supervision:** Katharine H. McVeigh.

**Writing – original draft:** Eugenie Poirot, Carrie W. Mills, Katharine H. McVeigh.

**Writing – review & editing:** Eugenie Poirot, Carrie W. Mills, Andrew D. Fair, Krishika A. Graham, Emily Martinez, Lauren Schreibstein, Achala Talati, Katharine H. McVeigh.

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
