## [Decision Letter · Decision Letter 0]

3 Mar 2020

PONE-D-20-00230

Evaluation of a health information exchange system for microcephaly case-finding — New York City, 2013—2015

PLOS ONE

Dear Ms. McVeigh,

Thank you for submitting your manuscript to PLOS ONE. After careful consideration, we feel that it has merit but does not fully meet PLOS ONE’s publication criteria as it currently stands. Therefore, we invite you to submit a revised version of the manuscript that addresses the points raised during the review process.

The Authors are expected to address the comments by all Reviewers. In particular, please provide more details on how the data were queried (Reviewers #1 and #3), how missing data was handled (Reviewer #2), differences between the two reporting systems, the analysis and state the study objective clearly (Reviewer #3).

We would appreciate receiving your revised manuscript by Apr 17 2020 11:59PM. To enhance the reproducibility of your results, we recommend that if applicable you deposit your laboratory protocols in protocols.io, where a protocol can be assigned its own identifier (DOI) such that it can be cited independently in the future. For instructions see: http://journals.plos.org/plosone/s/submission-guidelines#loc-laboratory-protocols

We look forward to receiving your revised manuscript.

Kind regards,

Eric HY Lau, Ph.D.

Academic Editor

PLOS ONE

Additional Editor Comments (if provided):

The Authors are expected to address the comments by all Reviewers. In particular, please provide more details on how the data were queried (Reviewers #1 and #3), how missing data was handled (Reviewer #2), differences between the two reporting systems, the analysis and state the study objective clearly (Reviewer #3).

Journal Requirements:

Please ensure that your manuscript meets PLOS ONE's style requirements, including those for file naming. The PLOS ONE style templates can be found at http://www.plosone.org/attachments/PLOSOne_formatting_sample_main_body.pdf and http://www.plosone.org/attachments/PLOSOne_formatting_sample_title_authors_affiliations.pdf

Reviewers' comments:

Reviewer's Responses to Questions

**Comments to the Author**

1. Is the manuscript technically sound, and do the data support the conclusions?

Reviewer #1: Yes

Reviewer #2: Yes

Reviewer #3: Partly

2. Has the statistical analysis been performed appropriately and rigorously? 

Reviewer #1: Yes

Reviewer #2: Yes

Reviewer #3: I Don't Know

3. Have the authors made all data underlying the findings in their manuscript fully available?

Reviewer #1: No

Reviewer #2: Yes

Reviewer #3: No

4. Is the manuscript presented in an intelligible fashion and written in standard English?

Reviewer #1: Yes

Reviewer #2: Yes

Reviewer #3: Yes

5. Review Comments to the Author

Reviewer #1: Thank you for this opportunity to review this paper. As someone who has worked at the federal, state, regional, and organization level on HIE, I found this very interesting. This manuscript was very well written, easy to read, and easy to follow. My minor comments are as follows:

--This may have more relevance in other surveillance situations, not just microcephaly. This should be stated.

--The methods sections was light. There needs to be more around how the data were queried

--Discussion and Conclusion. More details around the case that was missed (L146)

Reviewer #2: The study aims at evaluation of Health Information Exchange (HIE) data at three Bronx hospitals in New York City for surveillance and detection of microcephaly cases diagnosed at birth during Jan 1, 2013–Dec 31, 2015 before Zika virus introduction in North America. The use of HIE data along with other data sources for surveillance is a known practice. Methodologically the paper has shown little novelty. The domain, the application and results are interesting, although the sensitivity 58.21%, seems low. I’m also curious to know how the authors deal with missing data in the HIE and the chart reviews. Also, I suggest the authors to discuss potential biases in this study.

Minor comments:

Line 61: “these rapid and active surveillance components has” should be “these rapid and active surveillance components have”

Line 88: “was excluded from analysis” should be “was excluded from the analysis”

Line 103: “One of 4 cases meet the microcephaly case definition” should be “One of 4 cases meets the microcephaly case definition”

Line 166: “First, detection of” should be “First, the detection of”

Reviewer #3: The authors present an interesting analysis although the manuscript is vague and does not provide enough detail to properly evaluate the work.

Major comments

1) The objective of the study isn't well defined. Based on lines 73-74, the objective was to estimate how well the suspected case definition performs in terms of specificity and sensitivity?

2) The differences between the two reporting systems needs to be more clearly described. For example, routine reporting relies on a congenital malformation registry that is based on hospital reports (using ICD 10 codes) whereas the rapid reports are based on the same codes but are more frequently obtained from the same hospitals (I'm assuming that the ICD codes for Zika congenital syndrome has been updated in the routine reporting as well).

3) It also sounds like it was a capture-recapture approach that was taken for the study? If so, please further describe the approach as it will help the reader understand what has been done and the objective of the study. Nothing is mentioned at all about the analysis - how were the data obtained (in what format), what about basic characteristics of the suspected cases (any information on the mothers, when were they born, etc). Record linkage through what exactly? how were estimates of sensitivity, specificity, etc were estimated - using what type of regression? Sensitivity analysis based on the specificity and sensitivity of the case definitions should also be factored into the estimates.

4) Case definitions should be provided as supplemental information.

5) In terms of data availability, what data are available exactly and how (contact corresponding author)?

6. PLOS authors have the option to publish the peer review history of their article (what does this mean?). If published, this will include your full peer review and any attached files.

Reviewer #1: No

Reviewer #2: No

Reviewer #3: No

---

## [Author Response · Author response to Decision Letter 0]

21 Apr 2020

Dear Academic Editor: 

Thank you for inviting us to submit a revised version of our manuscript entitled “Evaluation of a health information exchange system for microcephaly case-finding — New York City, 2013—2015" to PLOS ONE. Following is a rebuttal letter that responds to each point raised by the academic editor and reviewers.

Review Comments to the Author

Academic Editor:

The Authors are expected to address the comments by all Reviewers. In particular, please provide more details on how the data were queried (Reviewers #1 and #3), how missing data was handled (Reviewer #2), differences between the two reporting systems, the analysis and state the study objective clearly (Reviewer #3).

Response: Thank you for the constructive feedback. Below the authors have responded and addressed the comments made by all reviewers with particular emphasis on providing more details on how the data were queried (Reviewers #1 and #3), how missing data was handled (Reviewer #2), differences between the two reporting systems, the analysis and study objective (Reviewer #3).

Reviewer #1: 

Comment 1. Thank you for this opportunity to review this paper. As someone who has worked at the federal, state, regional, and organization level on HIE, I found this very interesting. This manuscript was very well written, easy to read, and easy to follow. 

Response: Thank you – we are pleased you found the study to be interesting and clearly written. 

My minor comments are as follows:

Comment 2. This may have more relevance in other surveillance situations, not just microcephaly. This should be stated.

Response: Thank you for the comment. We completely agree. HIE systems present important opportunities to advance public health surveillance, in ways that can increase timeliness and completeness of public health surveillance. The literature has linked HIE systems to improvements in child and adolescent immunization status, timeliness of notifiable disease reporting, reductions in duplicative diagnostic testing and identification of drug seeking behaviors, and improved identification of high utilizing vulnerable patients returning within 72 hours of initial emergency department discharge. The Introduction of the manuscript now references the literature citing these examples where HIE systems have been used to enhance disease surveillance practices. 

In the discussion, the authors do describe scenarios where HIEs could be used to support newborn microcephaly surveillance. Additional language has now been added to clarify that these applications could similarly supplement traditional approaches to disease surveillance and facilitate surveillance for other conditions of public health interest, including emerging health threats. 

Comment 3. The methods sections was light. There needs to be more around how the data were queried.

Response: The authors have added more language to the methods around how the data were queried. The methods now further describe the HIE database, the approach taken to query and identify records meeting criteria, the data elements on suspected cases identified by HIE that were extracted for use in the match analysis. 

Comment 4. Discussion and Conclusion. More details around the case that was missed (L146)

Response: We agree with the reviewer that this case is perplexing. As we now indicate in our manuscript, we found no explanation for the missed case, but hypothesize that the diagnostic code was recorded in a secondary field in the EMR that was picked up by the HIE but not by the reporting hospital’s query in the NYSRCR study.

Reviewer #2: 

Comment 1. The study aims at evaluation of Health Information Exchange (HIE) data at three Bronx hospitals in New York City for surveillance and detection of microcephaly cases diagnosed at birth during Jan 1, 2013–Dec 31, 2015 before Zika virus introduction in North America. The use of HIE data along with other data sources for surveillance is a known practice. Methodologically the paper has shown little novelty. The domain, the application and results are interesting, although the sensitivity 58.21%, seems low. I’m also curious to know how the authors deal with missing data in the HIE and the chart reviews. Also, I suggest the authors to discuss potential biases in this study.

Response: In this analysis, the authors did not return to the charts the HIE missed to uncover the reasons why these cases were missed. However, the authors suspect it may be due to a lack of consistent and quality EMR documentation. The authors have now addressed this point in the limitations section of the Discussion. 

Minor comments:

Comment 2. Line 61: “these rapid and active surveillance components has” should be “these rapid and active surveillance components have”

Response: We agree. The phrase has been revised and now reads, “these rapid and active surveillance components have…”.

Comment 3. Line 88: “was excluded from analysis” should be “was excluded from the analysis”

Response: Revision made. The sentence now reads, “one suspected case born to a non-NYC resident mother was excluded from the analysis”. 

Comment 4. Line 103: “One of 4 cases meet the microcephaly case definition” should be “One of 4 cases meets the microcephaly case definition”

Response: Suggested change made. 

Comment 5. Line 166: “First, detection of” should be “First, the detection of”

Response: Change made. The sentence now reads, “First, the detection of cases using HIE data was limited to microcephaly diagnoses queried using a single ICD-CM-9 or ICD-CM-10 code on day of birth or during the first two days after birth”. 

Reviewer #3: The authors present an interesting analysis although the manuscript is vague and does not provide enough detail to properly evaluate the work.

Response: Thank you. We are pleased you found the analysis interesting. We have revised the manuscript to address the comments you outline below. 

Major comments:

Comment 1. The objective of the study isn't well defined. Based on lines 73-74, the objective was to estimate how well the suspected case definition performs in terms of specificity and sensitivity?

Response: We have revised the introduction to clarify the objective and rationale for our approach. The manuscript now reads, “The objective of this analysis was to determine the extent to which querying of HIE data could replicate the yield obtained by the NYSRCR study through more labor-intensive and costly approaches, involving direct inquiry of hospital records, querying administrative discharge data, and chart abstraction. The sensitivity, specificity, positive and negative predictive values of querying HIE data for identifying confirmed cases of microcephaly were estimated.” 

Comment 2. The differences between the two reporting systems needs to be more clearly described. For example, routine reporting relies on a congenital malformation registry that is based on hospital reports (using ICD 10 codes) whereas the rapid reports are based on the same codes but are more frequently obtained from the same hospitals (I'm assuming that the ICD codes for Zika congenital syndrome has been updated in the routine reporting as well).

Response: Thank you for the comment. In the manuscript, the authors describe 3 reporting systems: 1) routine reporting, 2) direct inquiry of hospitals followed by medical chart review, and 3) querying of HIE data. Routine reporting (#1) is introduced in the Introduction as relying on hospitals and providers to report individual cases to congenital malformations registries in accordance with state public health laws and regulations. However, the authors did not attempt to compare HIE to routine CMR reporting, which may have led to confusion by the reviewer. Surveillance of birth defects in New York conducted by the New York Congenital Malformations Registry receives reports from hospitals on major birth defects in infants and children diagnosed before the age of 2 years. Thus, routine CMR reporting is not helpful for outbreak detection or case-finding. 

The analysis described in this manuscript evaluated the extent to which querying of HIE data (#2) could replicate the yield obtained in the NYSRCR study through direct inquiry of hospital records (#3). Using the same criteria (diagnosis codes) and case definition as those used for the NYSRCR study, the authors evaluated the use of querying HIE data to replace direct inquiry of hospitals or enhance direct inquiry of hospitals by confirming cases identified through both systems and identifying additional cases not captured through hospital inquiry. HIE can replace or complement direct inquiry depending on resources available. We have revised the Methods to clarify the reporting systems being compared, and Discussion and Conclusion to more directly address these points and reduce confusion. 

Comment 3. It also sounds like it was a capture-recapture approach that was taken for the study? If so, please further describe the approach as it will help the reader understand what has been done and the objective of the study. Nothing is mentioned at all about the analysis - how were the data obtained (in what format), what about basic characteristics of the suspected cases (any information on the mothers, when were they born, etc). Record linkage through what exactly? how were estimates of sensitivity, specificity, etc were estimated - using what type of regression? Sensitivity analysis based on the specificity and sensitivity of the case definitions should also be factored into the estimates.

Response: Thank you for the comment. The authors did not design this as a capture-recapture study. The design of the analysis was a criterion-related validation study where direct inquiry of hospitals followed by medical chart review/abstraction was the “gold standard” model. The study design for this analysis was premised on the assumption that direct inquiry would capture 100% of cases. 

The revised manuscript expands on the methods of the analysis to address the concerns of the reviewer. Cases identified using the same microcephaly diagnostic codes born to NYC resident mothers were matched for three birth hospitals covered by both data sources using birth hospital name, date of birth, and medical record number. The select birth hospitals in the Bronx had cases in the NYSRCR that were also in the HIE system. Records were linked and matched using a deterministic approach. Cases were classified as a match if the two records agreed on all identifiers and a nonmatch if the two records disagreed on any of the identifiers. The definitions of sensitivity, specificity, negative predictive value, and positive predictive value are described. 

No regression analysis or adjustments to sensitivity and specificity estimates were made. This analysis did not attempt to evaluate the microcephaly case definitions developed by CDC and the National Birth Defects Prevention Network (NBDPN). Our goal was not to see whether the expanded list of ICD-10-CM codes used in Zika Birth Defects Surveillance (ZBDS) captures the entire population of children with congenital microcephaly. Rather, this analysis attempted to apply the same microcephaly diagnostic codes (ICD-9-CM or ICD-10-CM) to two data sources (captured via querying HIE data or through direct inquiry of hospital records) to see if the yield was comparable. A limitation we mention in the discussion is that the HIE diagnosis was required to occur in within 2 days of birth but could have been made at any time during the newborn stay for the NYSRCR study.

Comment 4. Case definitions should be provided as supplemental information.

Response: Thank you for the comment. We have clarified our descriptions of suspected and confirmed cases and now explain that NYSRCR records were defined as non-cases (that is, not confirmed) if they had been misclassified (e.g., macrocephaly, microcephalus) or because both a physician diagnosis and anthropometric information needed to accurately classify head circumference percentile were missing. In addition, we have referred readers to the below reference [8] for further information on NBDPN case definitions which is publicly available to interested readers: 

National Birth Defects Prevention Network (NBDPN). NBDPN abstractor’s instructions. Houston, TX: National Birth Defects Prevention Network; 2016. http://www.nbdpn.org/docs/NBDPN_Case_Definition-SurveillanceMicrocephaly2016Apr11.pdf

Comment 5. In terms of data availability, what data are available exactly and how (contact corresponding author)?

Response: Thank you for the inquiry. The authors have made available all the data required to replicate the analysis described in this manuscript. The data can be found in Table 1. For readers interested in the data used in the New York State Retrospective Chart Review study [Graham et al. 2017] and information abstracted from charts by trained clinicals, we would refer readers to contact the corresponding authors Deborah J. Fox (deb.fox@health.nyc.gov) and Krishika A. Graham (kgraham1@health.nyc.gov) for data requests.

---

## [Decision Letter · Decision Letter 1]

4 Jul 2020

PONE-D-20-00230R1

Evaluation of a health information exchange system for microcephaly case-finding — New York City, 2013—2015

PLOS ONE

Dear Dr. McVeigh,

Thank you for submitting your manuscript to PLOS ONE. After careful consideration, we feel that it has merit but does not fully meet PLOS ONE’s publication criteria as it currently stands. Therefore, we invite you to submit a revised version of the manuscript that addresses the points raised during the review process.

The Authors are expected to address the comments by Reviewers #3. In additional to these comments, please address:

Abstract, please add the 95% Cis for sensitivity, PPV and NPV.

We look forward to receiving your revised manuscript.

Kind regards,

Eric HY Lau, Ph.D.

Academic Editor

PLOS ONE

Additional Editor Comments (if provided):

The Authors are expected to address the comments by Reviewers #3. In additional to these comments, please address:

1. Abstract, please add the 95% Cis for sensitivity, PPV and NPV.

Reviewers' comments:

Reviewer's Responses to Questions

**Comments to the Author**

1. If the authors have adequately addressed your comments raised in a previous round of review and you feel that this manuscript is now acceptable for publication, you may indicate that here to bypass the “Comments to the Author” section, enter your conflict of interest statement in the “Confidential to Editor” section, and submit your "Accept" recommendation.

Reviewer #2: All comments have been addressed

Reviewer #3: All comments have been addressed

2. Is the manuscript technically sound, and do the data support the conclusions?

Reviewer #2: Yes

Reviewer #3: Yes

3. Has the statistical analysis been performed appropriately and rigorously? 

Reviewer #2: Yes

Reviewer #3: Yes

4. Have the authors made all data underlying the findings in their manuscript fully available?

Reviewer #2: Yes

Reviewer #3: Yes

5. Is the manuscript presented in an intelligible fashion and written in standard English?

Reviewer #2: Yes

Reviewer #3: Yes

6. Review Comments to the Author

Reviewer #2: The comments and concerns have been promptly addressed. The paper is acceptable and contributes to the field.

Reviewer #3: The authors did an impressive job at thoroughly addressing all of the comments. The manuscript is clear and the objectives and approach of the study are well described. I have only a few minor comments that need addressing:

- In the methods section, it should be stated how the 95% CIs were generated (for Table 2) as well as using a Kappa statistic

- Table 2 - don't include (2013-2015) in column heading

- Line 256 - Preferable to not start a sentence with And

- Discussion - An important limitation of using HIE, as mentioned, is the poor sensitivity. The alternative approach (starting line 195), would seem the most reasonable/main way forward, as HIE would need to be supplemented with chart reviews. I would suggest that this should be reworded in terms of alternative approach.

- Discussion - limitation #2 - this is an important point for recommendations. There needs to be better standardization of reporting across EMRs, which is likely the root of the problem. I would make an explicit recommendation here.

7. PLOS authors have the option to publish the peer review history of their article (what does this mean?). If published, this will include your full peer review and any attached files.

Reviewer #2: No

Reviewer #3: No

---

## [Author Response · Author response to Decision Letter 1]

14 Jul 2020

Dear Academic Editor: 

Thank you for inviting us to submit a revised version of our manuscript entitled “Evaluation of a health information exchange system for microcephaly case-finding — New York City, 2013—2015" to PLOS ONE. Following is a rebuttal letter that responds to each point raised by the academic editor and reviewers.

Review Comments to the Author

Academic Editor:

The Authors are expected to address the comments by Reviewers #3. In additional to these comments, please address:

1. Abstract, please add the 95% Cis for sensitivity, PPV and NPV.

Response: Thank you for the feedback. The authors have added the 95% CI for sensitivity, PPV, and NPV to the abstract. Additionally, the authors have responded and addressed the comments made by all Reviewer #3 below.

Review Comments to the Author:

Reviewer #2: The comments and concerns have been promptly addressed. The paper is acceptable and contributes to the field.

Response: Thank you – we are pleased you found that the revised manuscript acceptable for publication and a contribution to the field. 

Reviewer #3: The authors did an impressive job at thoroughly addressing all of the comments. The manuscript is clear and the objectives and approach of the study are well described. I have only a few minor comments that need addressing:

Response: Thank you for the constructive feedback. We have addressed and responded to the minor comments that need addressing.

Comment 1. In the methods section, it should be stated how the 95% CIs were generated (for Table 2) as well as using a Kappa statistic.

Response: That authors have now added a statement to specify that corresponding Clopper-Pearson exact 95% confidence intervals were estimated around the parameters generated in Table 2. Standard logit confidence intervals were initially used for the predictive values but have been modified to reflect more conservative Clopper-Pearson exact 95% confidence intervals. A sentence has also been added to the methods section to indicate that a kappa statistic was calculated to measure correspondence between the two systems. All statistical analyses were performed using SAS 9.4 (SAS Institute Inc, Cary, North Carolina) and can be replicated. The authors state this clearly in the methods section. 

Comment 2. Table 2 - don't include (2013-2015) in column heading

Response: The authors have removed ‘(2013-2015)’ from the column heading in Table 2. 

Comment 3. Line 256 - Preferable to not start a sentence with And

Response: The authors have revised the sentence. The sentence does not start with the word ‘And’ but now reads, “In fact, despite the increasing adoption and implementation of EMR systems, there has been a lack of standardization in design and structure, as well as in adoption of documentation workflows.”

Comment 4. Discussion - An important limitation of using HIE, as mentioned, is the poor sensitivity. The alternative approach (starting line 195), would seem the most reasonable/main way forward, as HIE would need to be supplemented with chart reviews. I would suggest that this should be reworded in terms of alternative approach.

Response: In the discussion, the authors do describe scenarios where HIEs could be used to support newborn microcephaly surveillance. The authors introduce the alternative approach of using HIE data to identify cases requiring confirmatory chart review in scenarios where direct inquiry is warranted. The authors have now reworked the Discussion to describe this alternative approach more clearly. The authors are more explicit about how adding automated HIE reporting to a request and review system could improve efficiency. For systems that do rely on hospital inquiry and confirmatory chart review, cases identified by both systems could bypass human review so that limited resources could be devoted to investigating cases that are only picked up by one source. 

Comment 5. Discussion - limitation #2 - this is an important point for recommendations. There needs to be better standardization of reporting across EMRs, which is likely the root of the problem. I would make an explicit recommendation here.

Response: We completely agree. We have made a recommendation for better standardization of reporting across EMRs to the Discussion. The manuscript now reads, “These variations call for better standardization of reporting and HIE practices across EMR systems. Continued effort to develop standards for EMR design that will produce clearer documentation of the clinical workflow and requirements for data capture could lessen these challenges and improve how EMR systems can be used for public health surveillance and assist health departments”.

---

## [Editor Report · Decision Letter 2]

27 Jul 2020

Evaluation of a health information exchange system for microcephaly case-finding — New York City, 2013—2015

PONE-D-20-00230R2

Dear Dr. McVeigh,

We’re pleased to inform you that your manuscript has been judged scientifically suitable for publication and will be formally accepted for publication once it meets all outstanding technical requirements.

Kind regards,

Eric HY Lau, Ph.D.

Academic Editor

PLOS ONE
---

## [Editor Report · Acceptance letter]

5 Aug 2020

PONE-D-20-00230R2 

Evaluation of a health information exchange system for microcephaly case-finding — New York City, 2013—2015 

Dear Dr. McVeigh:

I'm pleased to inform you that your manuscript has been deemed suitable for publication in PLOS ONE. Congratulations! Your manuscript is now with our production department. 

Kind regards, 

on behalf of

Dr. Eric HY Lau 

Academic Editor

PLOS ONE